# A Viable Alternative. Comment on Kohmer et al. Self-Collected Samples to Detect SARS-CoV-2: Direct Comparison of Saliva, Tongue Swab, Nasal Swab, Chewed Cotton Pads and Gargle Lavage. *J. Clin. Med.* 2021, *10*, 5751

**DOI:** 10.3390/jcm11154501

**Published:** 2022-08-02

**Authors:** Cristoforo Fabbris, Riccardo Camerotto, Veronica Battistuzzi, Giacomo Spinato

**Affiliations:** 1Department of Medicine, University of Padova, Via Giustiniani, 2, 35128 Padova, Italy; 2Otolaryngology Department, Hospital of Treviso, University of Padova, 31100 Treviso, Italy; riccardo.camerotto@gmail.com (R.C.); veronica.battistuzzi@aulss2.veneto.it (V.B.); 3Section of Otolaryngology, Regional Centre for Head and Neck Cancer, Department of Neurosciences, University of Padova, 31100 Treviso, Italy; giacomo.spinato@unipd.it

We read with great interest and would like to comment on the article “Self-Collected Samples to Detect SARS-CoV-2: Direct Comparison of Saliva, Tongue Swab, Nasal Swab, Chewed Cotton Pads and Gargle Lavage” [1]. In the pandemic era, early and reliable diagnosis is needed in order to reduce infections by as much as possible [2,3,4]. The authors performed a very interesting and useful study on different tools toward achieving this goal.

As can be seen in the text, there is not an ideal method to collect samples and obtain a diagnosis of Severe Acute Respiratory Syndrome Coronavirus 2 (SARS-CoV-2) infection, and we agree with this statement. According to the paper’s results, diagnostic sensitivity was 92.8% with saliva samples, 89.1% with gargling, 85.1% with nasal swabs, 74.2% with tongue swab, and 70.2% with saliva by chewing a cotton pad [1].

Since SARS-CoV-2 highly colonizes the nasopharynx, sampling methods should mainly reach this area [5]. Nasal lavages with saline solution are widely used at all ages as a prophylactic or curative strategy to improve nasal and upper airway symptoms [6]. They are a non-invasive method to clean nasal cavities and the nasopharynx (which can be also performed in children), but they can also be used to collect samples from the posterior area of the nose. This tool, indeed, has been effectively used in the past to detect upper airway inflammation, to assess the nasal response to irritants, and even to detect viral infections [2,4]. Therefore, they have been confirmed to be a valid alternative to traditional methods, especially in less compliant subjects.

As reported in the literature, nasal irrigation with isotonic saline solution has been proved to have 97.7% sensitivity and 98.9% accuracy [7]. Moreover, lavages can be easily self-performed, thus avoiding any need of protective equipment for healthcare professionals when collecting samples. Lavages have similar results to nasopharyngeal swabs, which is why nasal irrigation may represent one of the most viable tools. Moreover, it is important to note that this method is overall devoid of complications and is well tolerated [6].

Kohmer et al. [1] showed an interesting and correct statistical analysis, but there was no mention of specificity. We understand that the aim of the authors’ paper was to find a self-collection method for obtaining a sample to identify SARS-CoV-2, which is the main goal of diagnostic procedures. Additionally, in order to identify and isolate positive subjects to prevent infection in other subjects, an ideal method should also aim to not give false positives. According to the authors’ results, we can conclude that the different tools are able to avoid false negatives, but we cannot be sure regarding false positives. Nasal lavages, on the other hand, have been proved to have 100% specificity and 98.9% accuracy [7]. 

In conclusion, even if they are not a standardized diagnostic method, nasal irrigations have been proved to be a reliable way to obtain diagnosis, even more than the other procedures analyzed by the authors.

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
