# Peer review of "A Viable Alternative. Comment on Kohmer et al. Self-Collected Samples to Detect SARS-CoV-2: Direct Comparison of Saliva, Tongue Swab, Nasal Swab, Chewed Cotton Pads and Gargle Lavage. J. Clin. Med. 2021, 10, 5751"

_jcm, 2022, doi:10.3390/jcm11154501_

Round 1

Reviewer 1 Report

Fabbris et al. present a couple of comments on a report by Kohmer et al recently published on this journal on sampling methods for SARSCoV2 detection in various oropharyngeal compartments.

While both points have merit in theory, the commentary itself needs a lot of work to acquire significance:

1)     Nasopharyngeal lavage is not a new tool, having been utilized for detection of viruses in the upper respiratory tract since the late eighties, particularly in community based studies, and especially in early childhood, when swabbing might be inconvenient. As a rapid examination of the literature would confirm, adenovirus, RSV, rhinovirus  metapneumovirus, etc have been successfully detected in nasal/nasopharyngeal lavage fluids. It is appropriate that the commentary specify this, particularly because every single one of the references besides Kohmer et al is a self-cite.

2)     Yes, Kohmer et al do not offer specificity data. However, they are comparing different self-collection techniques to obtain specimens to be tested with the same assays, versus a gold standard of professionally collected nasopharyngeal swabs samples. In other words, the question is: can this or this other self-collection method provide a sample that allows to identify SARS-CoV2, in individuals with a positive nasopharyngeal swabs sample? For this reason, Kohmer et al do not include participants with a negative nasopharyngeal swabs result, and cannot calculate specificity. In the authors’ paper cited as reference 8, only participants with a positive rapid antigen test were recruited as “Cases” and only HCW whose nasopharyngeal swab tested negative were recruited as “Controls”. Specificity was found to be 100%, which is to be expected, but is also immaterial, as a negative swab and a positive lavage *tested with the same assays* would be considered indicative of infection (particularly in HCW in a  pandemic context) rather than a false positive.

Author Response

Dear Reviewer,

We are very grateful to you for your expert opinion and helpful comments. We have addressed these comments. Changes in the manuscript are highlighted in track canges mode and have outlined our response below.

We did our best to improve our manuscript as possible and hope now it’s suitable for publication.

1) We added some references regarding the use of nasal irrigations as a detection method, and deleted some less-useful references.

2) We edited tthe paragraph regarding specificity and left just a mention to this parameter.

Reviewer 2 Report

Comments from the authors are valuable. They pointed out that a considerable alternative sample collection method was ignored by Kohmer et. al. and the absence of specificity which was extremely important for COVID19 screening.

I suggest the contents in lines 50 to 55 and ref 8 be retained rather than deleted. 

Author Response

Dear Reviewer,

many thanks for your comments. We retained the lines you requested. Reference 8 was already retained and changed into reference 7.

We hope now the manuscript be suitable.

Best regards,

Cristoforo Fabbris (on behalf of all authors)